# RedMotion: Motion Prediction via Redundancy Reduction

**Royden Wagner**[1], **Ömer Şahin Taş**[2], **Marvin Klemp**[1], **Carlos Fernandez**[1], **Christoph Stiller**[1]
[1] *Karlsruhe Institute of Technology*
[2] *FZI Research Center for Information Technology*

**Reviewed on OpenReview:** *https://openreview.net/forum?id=xl1KhKT3Xx*

## Abstract

We introduce RedMotion, a transformer model for motion prediction in self-driving vehicles that learns environment representations via redundancy reduction. Our first type of redundancy reduction is induced by an internal transformer decoder and reduces a variable-sized set of local road environment tokens, representing road graphs and agent data, to a fixed-sized global embedding. The second type of redundancy reduction is obtained by self-supervised learning and applies the redundancy reduction principle to embeddings generated from augmented views of road environments. Our experiments reveal that our representation learning approach outperforms PreTraM, Traj-MAE, and GraphDINO in a semi-supervised setting. Moreover, RedMotion achieves competitive results compared to HPTR or MTR++ in the Waymo Motion Prediction Challenge. Our open-source implementation is available at: https://github.com/kit-mrt/future-motion

## 1 Introduction

It is essential for self-driving vehicles to understand the relation between the motion of traffic agents and the surrounding road environment. Motion prediction aims to predict the future trajectory of traffic agents based on past trajectories and the given traffic scenario. Recent state-of-the-art methods (e.g., Shi et al. (2022); Wang et al. (2023); Nayakanti et al. (2023)) are deep learning methods trained using supervised learning. As the performance of deep learning methods scales well with the amount of training data (Sun et al., 2017; Kaplan et al., 2020; Zhai et al., 2022), there is a great research interest in self-supervised learning methods, which generate supervisory signals from unlabeled data. While self-supervised methods are well established in the field of computer vision (e.g., Chen et al. (2020); Radford et al. (2021); He et al. (2020)), their application to motion prediction in self-driving has only recently started to emerge (e.g., Xu et al. (2022); Azevedo et al. (2022)). As in computer vision applications, generating trajectory datasets for self-driving can involve intensive post-processing, such as filtering false positive detections and trajectory smoothing.

We introduce RedMotion, a transformer model for motion prediction that incorporates two types of redundancy reduction for road environments. Specifically, our model learns augmentation-invariant features of road environments as self-supervised pre-training. We hypothesize that by using these features, relations in the road environment can be learned, providing important context for motion prediction.

We target transformer models for three reasons: **(a)** They are successfully applied to a wide range of applications in natural language processing (e.g., Vaswani et al. (2017); Brown et al. (2020); OpenAI (2023)), computer vision (e.g., Dosovitskiy et al. (2020); Carion et al. (2020); Meinhardt et al. (2022)), and time-series prediction (e.g., Zhou et al. (2021; 2022)). Therefore, it is likely that enhancements in training mechanisms in a particular application will also apply to other applications. **(b)** They have no inductive biases for generating features based on spatial correlations (Raghu et al., 2021). Therefore, appropriate mechanisms must be learned from data. **(c)** The performance of transformer models on various downstream tasks scales very well with datasets (Kaplan et al., 2020; Zhai et al., 2022).

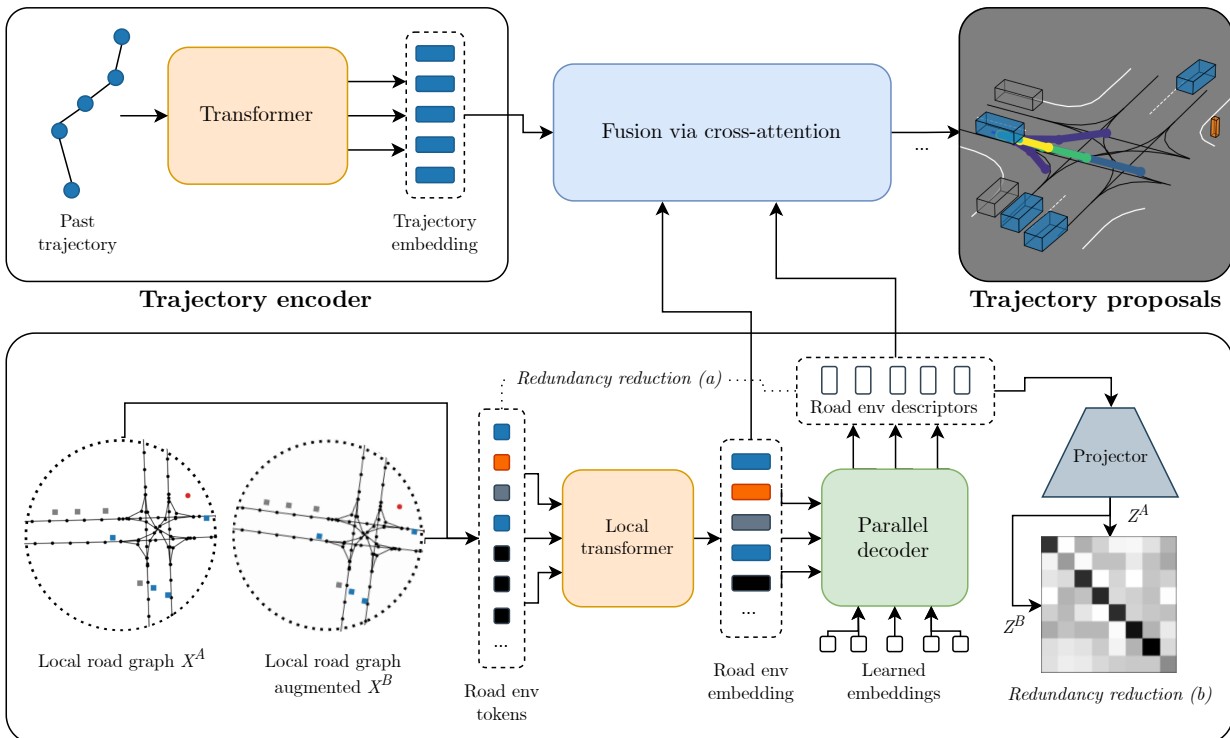

Figure 1: **RedMotion.** Our model consists of two encoders. The trajectory encoder generates an embedding for the past trajectory of the current agent. The road environment encoder generates sets of local and global road environment embeddings as context. We use two *redundancy reduction* mechanisms, *(a)* and *(b)*, to learn rich representations of road environments. All embeddings are fused via cross-attention to yield trajectory proposals per agent.

Our main contributions are the two types of redundancy reduction, which provide rich context representations for motion prediction:

1. An internal transformer decoder that reduces a variable-sized set of local road environment tokens to a fixed-sized global embedding (*Redundancy reduction (a)* in Figure 1).

2. Self-supervised redundancy reduction between embeddings generated from augmented views of road environments (*Redundancy reduction (b)* in Figure 1).

## 2 Related work

Recent works on motion prediction utilize a variety of deep learning models, including transformer models, graph-neural networks (GNNs), or convolutional neural networks (CNNs).

**Transformer models for motion prediction.** Ngiam et al. (2022) use PointNet (Qi et al., 2017) to encode polylines as a road graph. They use global scene-centric representations and fuse information from agent interactions across time steps and the road graph using self-attention mechanisms. Nayakanti et al. (2023) propose an encoder-decoder transformer model and study different attention (factorized and latent query (Jaegle et al., 2021)) and information fusion (early, late, and hierarchical) mechanisms. Zhang et al. (2023) propose a real-time capable transformer model that uses pairwise relative position encodings and an efficient attention mechanism based on the *k*-nearest neighbors algorithm.

**GNNs for motion prediction.** Gao et al. (2020) generate vectorized representations of HD maps and agent trajectories to use a fully-connected homogeneous graph. They learn a node embedding for every object in the scene in an agent-centric manner. Salzmann et al. (2020) use spatiotemporal graphs to jointly model scene context and interactions amongst traffic agents. Later works utilizing GNNs use heterogeneous graphs (Monninger et al., 2023; Grimm et al., 2023; Cui et al., 2023). Monninger et al. (2023) and Cui et al. (2023) highlight that choosing a fixed reference coordinate frame is vulnerable to domain shift and aim for viewpoint-invariant representations. These works store spatial information of lanes and agents in edges. Compared to transformer models, GNNs require additional modules to generate graph representations for multi-modal inputs. Furthermore, the scaling properties of GNNs can be undesirable.

**CNNs for motion prediction.** CNN-based approaches have emerged as straightforward yet effective baselines in motion prediction. Konev et al. (2022) use standard CNNs, with only the head being adapted to motion prediction. Girgis et al. (2022) process map information with a CNN and decode trajectory predictions with a transformer. However, CNNs typically tend to require larger models compared to GNNs or transformer models due to the low information content per pixel versus in vector representations.

**Self-supervised learning for motion prediction.** Labeled data requirements of preceding approaches motivate the application of self-supervised learning to motion prediction. Balestriero et al. (2023) categorize self-supervised representation learning methods into major families. *The deep metric learning family:* PreTraM (Xu et al., 2022) exploits for contrastive learning that a traffic agent's trajectory is correlated to the map. Inspired by CLIP (Radford et al., 2021), the similarity of embeddings generated from rasterized HD map images and past agent trajectories is maximized. Therefore, past trajectories are required, which limits the application of this method to annotated datasets. Ma et al. (2021) improve modeling interactions between traffic agents via contrastive learning with SimCLR (Chen et al., 2020). They rasterize images of intersecting agent trajectories and train the corresponding module by maximizing the similarity of different views of the same trajectory intersection. Accordingly, only a small part of the motion prediction pipeline is trained in a self-supervised manner and annotations are required to determine the trajectory intersections. *The masked sequence modeling family:* Chen et al. (2023) and Yang et al. (2023) propose masked autoencoding as pre-training for motion prediction. Inspired by masked autoencoders (He et al., 2022), they mask out parts of the road environment and/or past trajectory points and train to reconstruct them. When applied to past trajectory points, these approaches require annotated trajectory data. *The self-distillation family:* GraphDINO (Weis et al., 2023) is a self-supervision objective designed to learn rich representations of graph structures and thus can be applied to road graphs used for motion prediction. Following DINO (Caron et al., 2021), the learning objective is a self-distillation process between a teacher and a student model without using labels. Compared to the previously mentioned methods, self-distillation methods tend to require more hyperparameter tuning (e.g., loss temperatures or teacher weight updates). Besides these families, Azevedo et al. (2022) use graph representations of HD maps to generate possible traffic agent trajectories. Trajectories are generated based on synthetic speeds and the connectivity of the graph nodes. The pre-training objective is the same as for the subsequent fine-tuning: motion prediction. While this method is well adapted to motion prediction, it requires non-trivial modeling of agent positions and synthetic velocities when applied to non-annotated data.

# 3 Method

We present our method in two steps. First, we focus on learning representations of road environments, then how these representations are used for motion prediction.

## 3.1 Redundancy reduction for learning rich representations of road environments

We use the redundancy reduction principle (Barlow, 2001; Zbontar et al., 2021) to learn rich representations of road environments. Following Ulbrich et al. (2015), we define a road environment as lane network and traffic agent data. In the context of deep learning, Zbontar et al. (2021) define redundancy reduction as reducing redundant information between vector elements of embeddings. We implement two types of redundancy reduction:

**(a) Redundancy reduction from local to global token sets.** We reduce a variable length set of local road environment tokens to a fixed-sized set of global road environment descriptors (RED). The set of local road environment tokens is generated via a local attention (Beltagy et al., 2020; Jiang et al., 2023) (see local self-attention in Figure 2). Therefore, these tokens represent lane and agent features within limited areas. To capture global context, we use a global cross-attention mechanism between a fixed-sized set of learned RED tokens and road environment tokens (see parallel decoder in Figure 1). In contrast to latent query attention (Jaegle et al., 2021; Nayakanti et al., 2023), RED tokens are used to learn global representations, while local attention is used to reduce the computational cost of processing long input sequences.

**(b) Redundancy reduction between global embeddings.** These embeddings are generated from RED tokens (see projector in Figure 1). This self-supervision objective, Road Barlow Twins (RBT), is based on Barlow Twins (Zbontar et al., 2021), which aims to learn augmentation-invariant features via redundancy reduction. For each training sample ($X^A$ in Figure 1) we generate an augmented view $X^B$. We use uniform distributions to sample random rotation (max. +/- 10°) and shift augmentations (max. +/- 1m). Afterwards, the local transformer and parallel decoder within the road environment encoder generate a set of RED tokens per input view. Finally, an MLP-based projector generates two embeddings ($Z^A$ and $Z^B$) from the RED tokens. For the two embedding vectors, a cross-correlation matrix is created. The training objective is to approximate this cross-correlation matrix to the corresponding identity matrix, while reducing the redundancy between individual vector elements. By approximating the identity matrix, similar RED token sets are learned for similar road environments. The redundancy reduction mechanism prevents that multiple RED tokens learn similar features of an environment, increasing expressiveness. Accordingly, our proposed self-supervision objective belongs to the similarity learning (i.e., canonical correlation analysis) family (Balestriero et al., 2023). The mentioned modules of the road environment encoder are shown with more details in Figure 2.

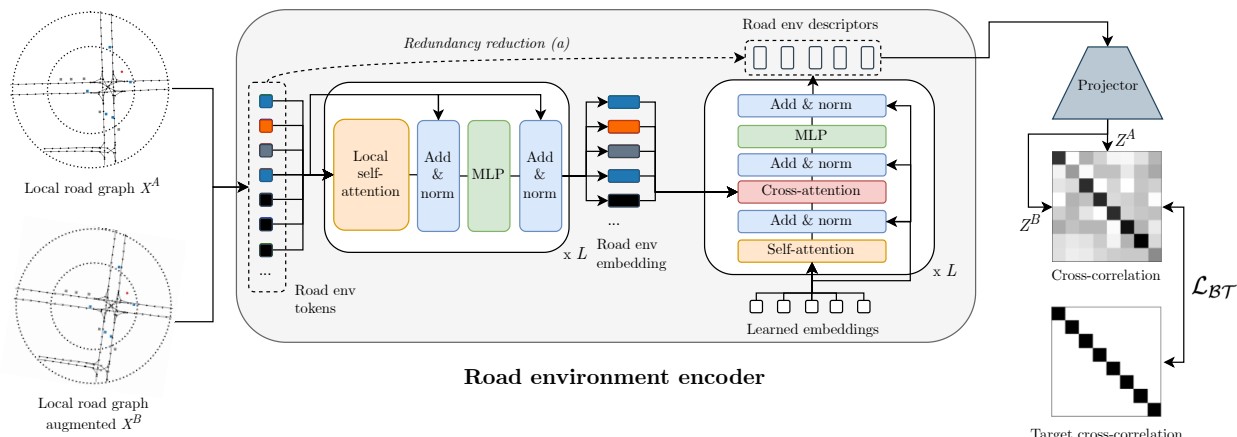

Figure 2: **Road environment encoder.** The circles in the local road graphs denote the maximum distance for considered lane network (outer) and agent nodes (inner). $\mathcal{L}_{\mathcal{BT}}$ is the Barlow Twins loss, $L$ is the number of modules.

## 3.2 Road environment description and motion prediction model

Our proposed motion prediction model (RedMotion) is a transformer model that builds upon the aforementioned two types of redundancy reduction. As input, we generate road environment tokens for agents and lanes based on a local road graph (see Figure 1). In this local road graph, the current agent is in the center and we consider lane nodes within a radius of 50 meters and agent nodes within 25 meters. As input for the road environment encoder, we use a list of these tokens sorted by token type, polyline, and distance to the current agent. The road environment encoder learns a semantic embedding for each token type, which is concatenated with the position relative to the current agent. We learn separate embeddings for static and

dynamic agents by using the current speed and the threshold of $0.0\,\mathrm{m/s}$ for classification. In Figure 3b static vehicles are marked as grey squares and dynamic vehicles as blue squares.

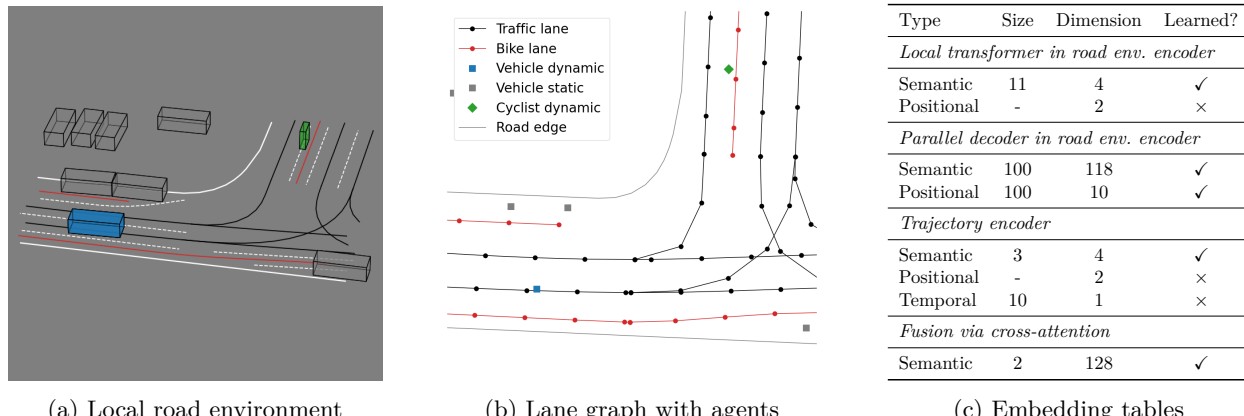

| (a) Local road environment | (b) Lane graph with agents | (c) Embedding tables |
| --- | --- | --- |

Figure 3: **Road environment description.** Local road environments are first represented as lane graphs with agents, afterwards, we generate token sets as inputs by using embedding tables for semantic types and temporal context. Positions are encoded relative to the current agent, except for RED tokens, which contain learned positional embeddings.

Figure 3c shows the vocabulary size (number of individual embeddings), the dimension of individual embeddings, and whether they are learned during training. In the road environment encoder, we additionally use a learned linear projection to project the concatenated semantic and positional embedding to the model dimension.

Since local relations are especially important for processing lanes, we use local windowed attention (Beltagy et al., 2020) layers. Compared to classical attention layers, these have a local attention mechanism within a limited window instead of a global one. In addition to the focus on local relations, this reduces memory requirements and allows us to process long input sequences (default: 1200 tokens). We use multiple stacked local attention layers in our road environment encoder (see Figure 2). Correspondingly, the receptive field of each token grows with an increasing number of local attention layers. Figure 4 shows how a polyline-like representation is built up for a traffic lane token (shown as black lines connecting lane nodes). For a better illustration, we show an example of an attention window of 2 tokens per layer. In our model, we use an attention window size of 16 tokens. The polyline-like representations built in the local transformer are set in relation to all surrounding tokens in the parallel decoder of RED tokens (shown as dashed grey lines in Figure 4). This is implemented by a global cross-attention mechanism from RED tokens to road env tokens (see Figure 1). Thus, global representation can be learned by RED tokens.

In addition to the road environment, we encode the past trajectory of the current agent with a standard transformer encoder (trajectory encoder in Figure 1). The trajectory encoder learns a semantic embedding per agent type, which is concatenated with both a temporal encoding and the relative position w.r.t. the current position. As temporal encoding, we use the number of time steps between the encoded time step and the time step at which the prediction starts. The concatenated embedding is then projected to the model dimension using a learned linear projection.

Afterwards, we fuse the embedding of the past trajectory with local and global (RED) road environment embeddings in two steps. First, we use regular cross-attention to fuse the past trajectory tokens with local road environment tokens. Second, we use a memory efficient implementation of cross-attention (Chen et al., 2021a) to add information from our global RED tokens (as shown in Figure 5). We concatenate both input sequences with learned fusion tokens ([Fusion]). We use the local fusion token as queries vector and concatenate it with our RED tokens to generate keys and values matrices for a standard attention module. Hence, the attention module computes: $\mathrm{softmax}(\frac{QK^{\intercal}}{\sqrt{d_K}}) \cdot V$, where $Q$, $K$, and $V$ are query, key, and value matrices, and $d_K$ is the dimension of key vectors. Then, we add the local fusion token to the attention output to generate a local-global fusion token. In Figure 5 we focus on the fusion mechanism of the local

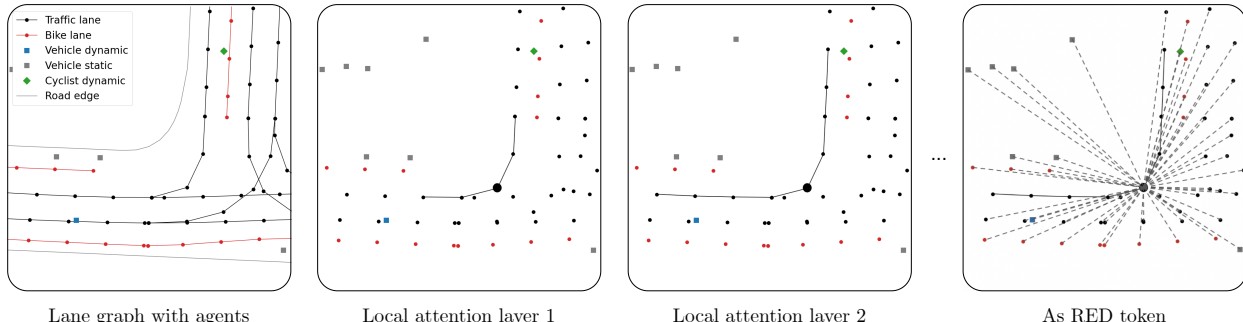

Figure 4: **Receptive field of a traffic lane token.** It expands in subsequent local attention layers, thereby enabling the token to gather information from related tokens within a larger surrounding area. Consequently, the road environment tokens initially form a disconnected graph and gradually transform into a fully connected graph. Best viewed, zoomed in.

fusion token with our RED tokens (local-global fusion). We proceed analogously when fusing the global fusion token with trajectory and local environment tokens (global-local fusion), which is shown as dashed line connecting the global fusion token to the output sequence. In this case, we replace the local fusion token with the global fusion token and RED tokens with trajectory and local environment tokens during fusion. For both ways, we compute this cross-attention mechanism in a token-to-sequence manner. This reduces the computational complexity compared to regular sequence-to-sequence attention from $O(n^2)$ to $O(2n)$, where $n$ is the sequence length. Finally, the output sequence is generated by concatenating the fused tokens with trajectory and local environment tokens.

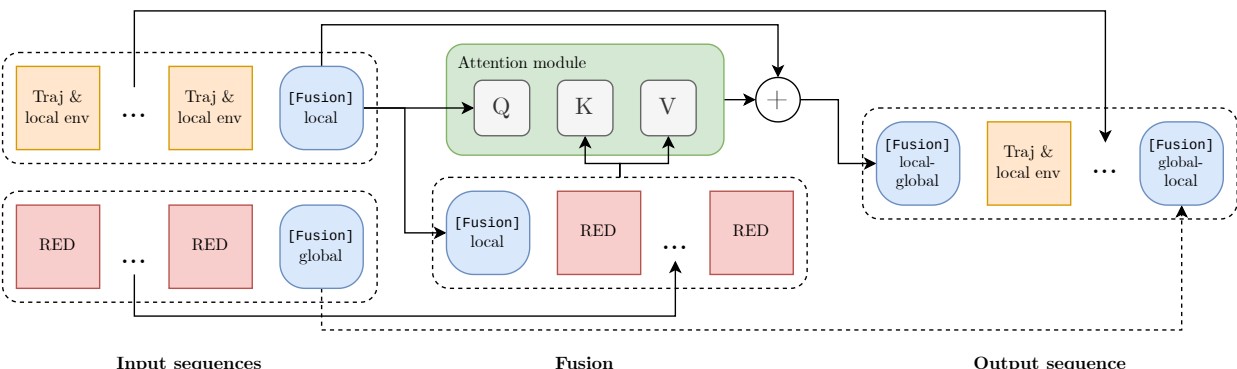

Figure 5: **Efficient cross-attention for feature fusion.** The output sequence contains features from the past trajectory, the local road environment, and global RED tokens.

After the fusion step, we use global average pooling over the set dimension to reduce the input dimension for an MLP-based motion decoder. This decoder regresses a configurable number of trajectory proposals and the corresponding confidences per agent. During inference, we use an agent-centric representation of the environment and predict trajectories marginally. In detail, each agent in the scene becomes the agent in the center when we predict his future trajectory.

## 4 Experiments

### 4.1 Comparing pre-training methods for motion prediction

In this set of experiments, we compare our representation learning approach with approaches from all other major families of self-supervised learning methods (Balestriero et al., 2023). Specifically, we evaluate against

contrastive learning with PreTraM (Xu et al., 2022), self-distillation using GraphDINO (Weis et al., 2023), and masked sequence modeling with Traj-MAE (Chen et al., 2023). Since self-supervised learning methods for motion prediction are only recently being developed, there are no common baseline models to compare them. Therefore, we use a modified version of our proposed model as baseline. Learning global environment features represented as RED tokens is our contribution, hence we remove the parallel decoder in the road environment encoder for the baseline version (see Figure 1). To allow a fair comparison, we increase the token dimension for the baseline from 128 to 192 to give both models a similar capacity (RedMotion baseline 9.9M params vs. RedMotion 9.2M params).

**Datasets.** We use the official training and validation splits of the Waymo Open Motion dataset (Ettinger et al., 2021) version 1.0 and the Argoverse 2 Forecasting dataset (Wilson et al., 2021) as training and validation data. Since pre-training is particularly useful when little annotated data is available, we use 100% of the training data for pre-training and fine-tune on only 12.5%, following common practice in self-supervised learning (Balestriero et al., 2023). In addition to the road environment, we use the trajectories of all traffic agents from the last second (Waymo) and last 5 seconds (Argoverse) as input during fine-tuning for motion prediction. The datasets are sampled with 10 Hz, accordingly we use 10 and 50 past time steps as input.

**Evaluation metrics.** We use the L5Kit (Houston et al., 2021) to evaluate the multimodal trajectory predictions of our models. Following common practice (Houston et al., 2021; Ettinger et al., 2021), we use the final displacement error (FDE) and the average displacement error (ADE) to evaluate trajectory proposals. These scores are evaluated in the oracle/minimum mode. Accordingly, the distance errors of the trajectory proposal closest to the ground truth are measured. For the Waymo dataset, minADE and minFDE metrics are computed at different prediction horizons of 3s and 5s, and averaged. For the Argoverse dataset, minADE and minFDE metrics are computed for the prediction horizon of 6s. Additionally, we compute the relative performance deltas w.r.t. the baseline model with $\Delta_{\text{rel}} = \frac{\text{minADE}_{\text{pre}} - \text{minADE}_{\text{base}}}{\text{minADE}_{\text{base}}} \cdot 100$, where $\text{minADE}_{\text{pre}}$ is the minADE score achieved with pre-training, whereas $\text{minADE}_{\text{base}}$ without pre-training. We compute all metrics considering 6 trajectory proposals per agent.

**Experimental setup.** For PreTraM, GraphDINO, and our pre-training method, we use the same augmentations described in Section 3. For Traj-MAE pre-training, we mask 60% of the road environment tokens and train to reconstruct them. For all methods, we train the pre-training objectives using our local road environment tokens, so that the lane network and social context are included. For PreTraM, we evaluate two configurations, map contrastive learning (MCL) and trajectory-map contrastive learning (TMCL). For MCL, the similarity of augmented views of road environments is maximized. For TMCL, the similarity of embeddings from road environments and past agent trajectories of the same scene is maximized.

We evaluate our method with four different configurations of redundancy reduction: with mean feature aggregation (mean-ag), with learned feature aggregation (learned-ag), with reconstruction (red-mae), and between environment and past trajectory embeddings (env-traj). Mean-ag refers to using the mean of the RED tokens as input to the projector in Figure 1. For learned-ag, we use an additional transformer encoder layer to reduce the dimension of RED tokens to 16 and concatenate them as input for the reduction projector. The red-mae configuration is inspired by masked sequence modeling and a form of redundancy reduction via reconstruction. In detail, we generate two views ($X^A$ and $X^B$) of road environments, randomly mask 60% of their tokens, and reconstruct $X^A$ from the masked version of $X^B$ and vice versa. Since we reconstruct cross-wise, the similarity between embedding representations of the augmented views is maximized during pre-training. The env-traj configuration is inspired by TMCL and reduces the redundancy between embeddings of past agent trajectories and RED tokens. Therefore, this configuration is inherently cross-modal but requires annotations of past agent trajectories.

For pre-training and fine-tuning, we use AdamW (Loshchilov & Hutter, 2019) as the optimizer. The initial learning rate is set to $10^{-4}$ and reduced to $10^{-6}$ using a cosine annealing learning rate scheduler (Loshchilov & Hutter, 2016). We pre-train and fine-tune all configurations for 4 hours and 8 hours using data-parallel training on 4 A100 GPUs. Following Konev et al. (2022), we minimize the negative multivariate log-likelihood loss for fine-tuning on motion prediction.

**Results.** Table 1 shows the results of this experiment. Overall, all pre-training methods improve the prediction accuracy in terms of minFDE and minADE. For our baseline model, our redundancy reduction

| Dataset | Model | Pre-training | Config | $minFDE_6$ $\downarrow$ | | $minADE_6$ $\downarrow$ | |
|---|---|---|---|---|---|---|---|
| Waymo | Baseline | None | | $1.371 \pm 0.018$ | | $0.670 \pm 0.009$ | |
| | | Traj-MAE* (Chen et al., 2023) | | $1.154 \pm 0.002$ | **-15.8%** | $0.542 \pm 0.001$ | -19.1% |
| | | PreTraM (Xu et al., 2022) | MCL | $1.250 \pm 0.012$ | -8.8% | $0.576 \pm 0.004$ | -14.0% |
| | | PreTraM (Xu et al., 2022) | TMCL* | $1.154 \pm 0.001$ | **-15.8%** | $0.525 \pm 0.001$ | **-21.6%** |
| | | GraphDINO (Weis et al., 2023) | | $1.257 \pm 0.010$ | -8.3% | $0.586 \pm 0.003$ | -12.5% |
| | | RBT (ours) | mean-ag | $1.159 \pm 0.006$ | -15.5% | $0.557 \pm 0.002$ | -16.9% |
| | RedMotion | RBT (ours) | mean-ag | $1.111 \pm 0.002$ | -19.0% | $0.568 \pm 0.001$ | -15.2% |
| | | RBT (ours) | learned-ag | $1.098 \pm 0.001$ | -19.9% | $0.555 \pm 0.001$ | -17.2% |
| | | RBT (ours) | red-mae | $1.092 \pm 0.002$ | -20.4% | $0.557 \pm 0.001$ | -16.9% |
| | | RBT (ours) | env-traj* | $1.058 \pm 0.009$ | **-22.8%** | $0.529 \pm 0.004$ | **-21.0%** |
| Argo 2 | RedMotion | None | | $2.353 \pm 0.024$ | | $1.157 \pm 0.004$ | |
| | | RBT (ours) | mean-ag | $2.285 \pm 0.010$ | -2.9% | $1.140 \pm 0.003$ | -1.5% |
| | | RBT (ours) | env-traj* | $2.265 \pm 0.020$ | **-3.7%** | $1.106 \pm 0.008$ | **-4.4%** |

Table 1: **Comparing pre-training methods for motion prediction.** Best scores are **bold**, second best are underlined. We evaluate on the Waymo Open Motion (Waymo) and the Argoverse 2 Forecasting (Argo 2) datasets and report mean ± standard deviation of 3 replicas per method and configuration. All methods are pre-trained on 100% and fine-tuned on 12.5% of the training sets. *Denotes methods that require past trajectory annotations.

reduction mechanism (b) (see Section 3) ranks second for the minFDE metric, marginally behind Traj-MAE and PreTraM in its TMCL configuration (only 0.3% worse). In terms of minADE, our mechanism ranks third behind Traj-MAE and PreTraM-TMCL. However, our mechanism is much less complex and has less data requirements. Compared to Traj-MAE, no random masking and no complex reconstruction decoder (transformer model) are required. Compared to PreTraM-TMCL, no past trajectory data is required.

When comparing to methods with similar requirements, our method outperforms PreTraM-MCL (-8.8% vs. -15.5% in minFDE) and GraphDINO (-8.3% vs. -15.5% in minFDE). For PreTraM-MCL, the question arises: "What is a good negative road environment?" Road environments of agents close to each other (e.g., a group of pedestrians) are much more similar than, for example, images of different classes in ImageNet (e.g., cars and birds). During contrastive pre-training, all samples in a batch other than the current one are treated as negative examples. Therefore, the pre-training objective becomes to learn dissimilar embeddings for rather similar samples. For GraphDINO, we hypothesize that more hyperparameter tuning could further improve the performance (e.g., loss temperatures or teacher weight update decay). When we combine our two redundancy reduction mechanisms a and b (lower group in Table 1), our RedMotion model outperforms all related methods by at least 4% in minFDE and achieves similar performance in the minADE metric. We hypothesize that the reason for the comparable worse performance in the minADE score is our trajectory decoding mechanism. Our MLP-based motion head regresses all points in a trajectory at once, thus individual points in a predicted trajectory are less dependent on each other than in recurrent decoding mechanisms. When fine-tuning, the error for the final trajectory point is likely to be higher than for earlier points and our model can learn to focus more on minimizing this loss term. Therefore, if pre-training improves the learning behavior of our model, this will affect the minFDE score more.

When comparing different configurations of our combined redundancy reduction objective (middle block in Table 1), the env-traj configuration performs best and the mean-ag configuration performs worst. However, similar to PreTraM-TMCL our env-traj configuration learns to map corresponding environment embeddings and past trajectory embeddings close to each other in a shared embedding space. Therefore, past trajectory data is required, which makes this objective less self-supervised and rather an improvement in data (utilization) efficiency. The learned-ag and red-mae configurations perform both better than the mean-ag configuration (1% improvement in minFDE and 2% improvement in minADE) and rather similar to each other. Since the learned-ag configuration has a lower computational complexity (no transformer-based reconstruction decoder but a simple MLP projector), we choose this pre-training configuration in the following.

On the Argoverse dataset (lower block in Table 1), we compare our RedMotion model without pre-training versus with our mean-ag and env-traj pre-training configurations. Our mean-ag configurations improves the

minFDE score by 2.9% and our env-traj configurations by 3.7%. This shows that our pre-training methods improve the prediction accuracy on both datasets. Overall, the achieved displacement errors are higher for the Argoverse dataset than for the Waymo dataset, since the metrics are not averaged over multiple prediction horizons, and likely because the Argoverse training split is smaller (about 1M vs. 2M agent-centric samples).

Figure 6 shows some qualitative results of our RedMotion model. Further qualitative results can be found in Appendix A.

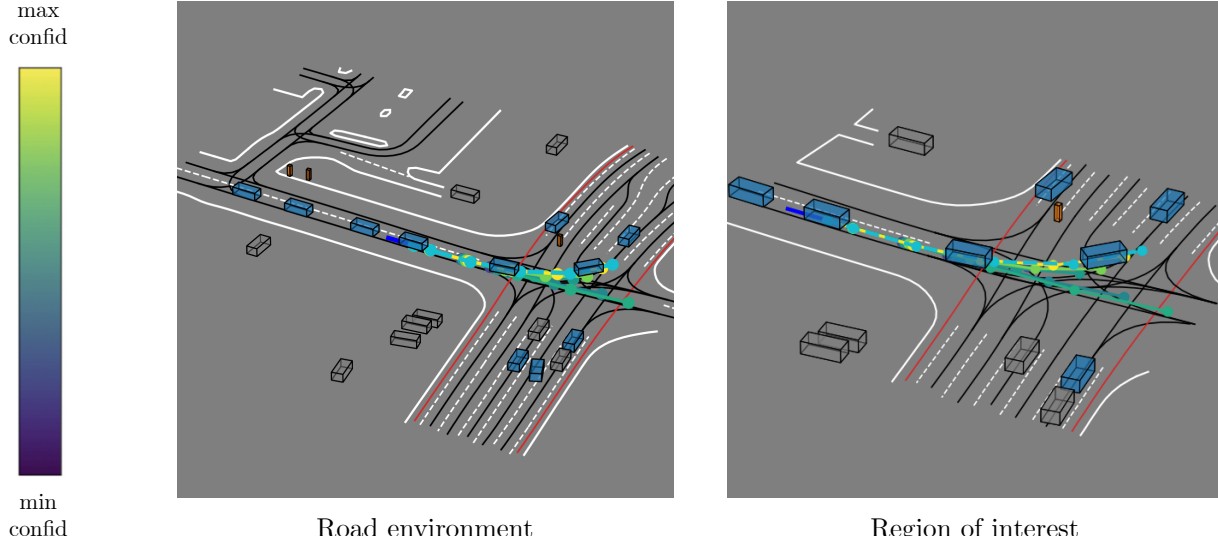

Road environment                   Region of interest

Figure 6: **Vehicle motion predictions.** Dynamic vehicles are marked as blue boxes, pedestrians as orange boxes, cyclists as green boxes, and static agents as grey boxes. Road markings are shown in white, traffic lane centerlines are black lines, and bike lane centerlines are red lines. The past trajectory of the ego agent is a dark blue line. The ground truth trajectory is cyan blue, the predicted trajectories are color-coded based on the associated confidence score using the viridis colormap on the left.

## 4.2 Comparing motion prediction models

We compare our RedMotion model with other recent models for marginal motion prediction on the Waymo Motion Prediction Challenge. Following the evaluation protocol of Zhang et al. (2023), we compare our methods against real-time capable methods without extensive post-processing (e.g., MTR adv-ens aggregates 6 trajectory proposals from an ensemble of 7 models with 64 proposals per model for each agent). However, we show results of non-real-time capable ensembles for reference.

As described in Section 4.1, our model with a basic MLP-based motion decoder (mlp-dec configuration) tends to focus more on later than on earlier trajectory points, which worsens minADE and mAP scores. Therefore, we additionally train a version of our model with a transformer decoder (tra-dec) as motion decoder, which is common amongst recent related methods (Girgis et al., 2022; Nayakanti et al., 2023; Zhang et al., 2023). In detail, we use a decoder with learned query tokens, which are transformed into trajectory proposals via attending to fused trajectory and road environment embeddings. For both variants, we use our redundancy reduction mechanisms to learn road environment embeddings. As embedding aggregation method, we use the learned-ag configuration from Section 4.1. As post-processing, our method does not require trajectory aggregation. Thus, we follow Konev (2022) and modify only the predicted confidences to improve mAP scores.

**Dataset.** We use 100% of the Waymo Open Motion training set for training to compare the performance of our model with that of other recent models. We perform evaluation on the validation and test splits.

**Evaluation metrics.** We use the same metrics for trajectories as in the previous set of experiments. However, this time, as in the Waymo Open Motion Challenge (Ettinger et al., 2021), the minFDE and minADE scores are computed by averaging the scores for the three prediction horizons of 3s, 5s, and 8s. Additionally, we report the minFDE and minADE scores for the prediction horizon of 8s. For the test split, we also report this years challenge main metric, the Soft mAP score. Following the current challenge rules, we compute the metrics for 6 trajectory proposals per agent.

**Results.** Table 2 shows the performance of our model in comparison to other motion prediction models. On the validation split, our model with a basic MLP-based motion decoder achieves the second lowest $\text{minFDE}_6$@8s score. Our model with a transformer decoder as motion decoder achieves the second best scores for the $\text{minFDE}_6$, $\text{minADE}_6$, and $\text{minADE}_6$@8s metrics. This shows that a transformer decoder adds modeling capacity and prevents our model from focusing too much on the final trajectory points during training. On the test spilt, our model with a transformer decoder ranks second in terms of $\text{minADE}_6$ and $\text{minADE}_6$@8s . For the main challenge metric, the Soft $\text{mAP}_6$ score, our model outperforms all comparable models. For reference, methods that employ ensembling achieve higher mAP scores yet comparable minADE and minFDE scores to our method. Therefore, they match our performance on average and are marginally better in assigning confidence scores, which likely stems from their extensive post-processing.

| Split | Method | Config | $\text{minFDE}_6 \downarrow$ | $\text{minADE}_6 \downarrow$ | $\text{minFDE}_6@8s \downarrow$ | $\text{minADE}_6@8s \downarrow$ | $\text{Soft mAP}_6 \uparrow$ |
|---|---|---|---|---|---|---|---|
| Val | MotionCNN (Konev et al., 2022) | ResNet-18 | 1.640 | 0.815 | - | - | - |
| | MotionCNN (Konev et al., 2022) | Xeption71 | 1.496 | 0.738 | - | - | - |
| | MultiPath++ (Varadarajan et al., 2022) | | - | - | 2.305 | 0.978 | - |
| | Scene Transformer (Ngiam et al., 2022) | marginal | 1.220 | 0.613 | 2.070 | 0.970 | - |
| | HPTR (Zhang et al., 2023) | | **1.092** | **0.538** | **1.877** | **0.874** | - |
| | RedMotion (ours) | mlp-dec | 1.271 | 0.701 | 1.952 | 1.110 | - |
| | RedMotion (ours) | tra-dec | 1.137 | 0.550 | 1.987 | 0.901 | - |
| | MTR* (Shi et al., 2022) | | 1.225 | 0.605 | - | - | - |
| | MTR++* (Shi et al., 2024) | | 1.199 | 0.591 | - | - | - |
| Test | MotionCNN (Konev et al., 2022) | Xeption71 | 1.494 | 0.740 | - | - | - |
| | Scene Transformer (Ngiam et al., 2022) | marginal | 1.212 | 0.612 | 2.053 | 0.980 | - |
| | HDGT (Jia et al., 2023) | | **1.107** | 0.768 | **1.898** | 1.284 | 0.371 |
| | MPA (Konev, 2022) | | 1.251 | 0.591 | 2.202 | 0.981 | 0.393 |
| | HPTR (Zhang et al., 2023) | | 1.139 | **0.557** | 1.954 | **0.910** | 0.397 |
| | RedMotion (ours) | tra-dec | 1.165 | 0.564 | 2.024 | 0.925 | **0.401** |
| | MTR* (Shi et al., 2022) | | 1.221 | 0.605 | 2.067 | 0.983 | 0.422 |
| | MTR++* (Shi et al., 2024) | | 1.194 | 0.591 | 2.024 | 0.961 | 0.433 |
| | Wayformer** (Nayakanti et al., 2023) | multi-axis | 1.128 | 0.545 | 1.942 | 0.892 | 0.434 |
| | MTR** (Shi et al., 2022) | adv-ens | 1.134 | 0.564 | 1.917 | 0.915 | 0.459 |

Table 2: **Comparing motion prediction models.** Best scores are **bold**, second best are underlined. *Denotes methods that require trajectory aggregation as post-processing. **Denotes methods that employ ensembling.

# 5 Contribution and future work

In this work, we introduced a novel transformer model for motion prediction in the field of self-driving. Our proposed model incorporates two types of redundancy reduction, an architecture-induced reduction and a self-supervision objective for augmented views of road environments. Our evaluations indicate that this pre-training method can improve the accuracy of motion prediction and outperform contrastive learning, self-distillation, and autoencoding approaches. The corresponding RedMotion model attains results that are competitive with those of state-of-the-art methods for motion prediction. Our method for creating global RED tokens provides a universal approach to perform redundancy reduction, transforming local context of variable length into a fixed-sized global embedding. In future work, this approach can be applied to other context representations, extending to further multi-modal inputs beyond agent and environment data.

**Acknowledgments**

We acknowledge the financial support by the German Federal Ministry of Education and Research (BMBF) within the project HAIBrid (FKZ 01IS21096A) and the German Federal Ministry for Economic Affairs and Climate Action (BMWK) within the project nxtAIM - Next Generation AI Methods.

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

# Appendix

## A Additional qualitative results

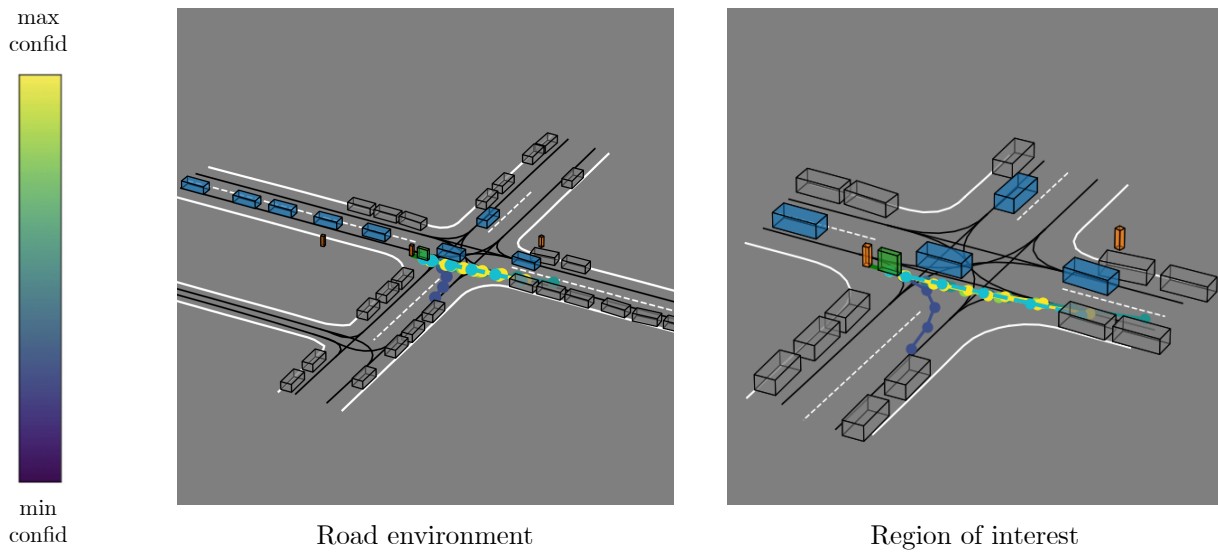

Figure 7: **Cyclist motion predictions.** We use the same color-coding as in Figure 6. This plot shows an error case as the blueish trajectory pointing downwards enters the inbound lane.

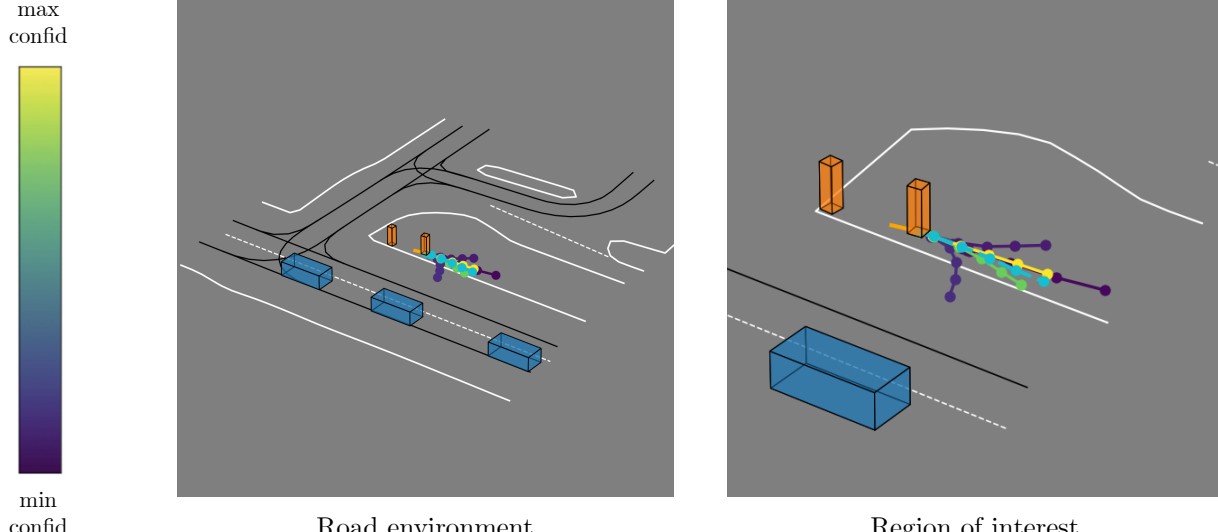

Figure 8: **Pedestrian motion predictions.** We use the same color-coding as in Figure 6.

## B    Limitations

We use a world on rails assumption (Chen et al., 2021b) for our model and predict trajectories marginally. Specifically, we predict the trajectory of each agent in a scene individually only considering the current state but not the predicted trajectories of the surrounding agents. Therefore, we can not enable consistency across all predictions in a scene as in joint motion prediction approaches (e.g., Ngiam et al. (2022)). Furthermore, we do not evaluate the robustness of our method w.r.t. out-of-distribution samples (e.g., Itkina & Kochenderfer (2023)). Finally, our pre-training method is not as label-efficient as self-supervised methods in computer vision (e.g., Chen et al. (2020)) that exclusively learn from raw data (i.e., unlabeled images). Our pre-training does not require motion data, but leverages the current state of agents and map data.

## C    Increasing the number of trajectory proposals

In this experiment, we increase the number of predicted trajectory proposals in our RedMotion model with a basic MLP-based motion decoder. Figure 9 shows the results on the validation split of the Waymo Open Motion dataset. Increasing the number of proposals from 6 to 16 decreases the minFDE score from 1.271 to 0.912 meters and the minFDE @8s score from 1.952 to 1.387 meters. This shows that our models performance scales well with the amount of trajectory proposals.

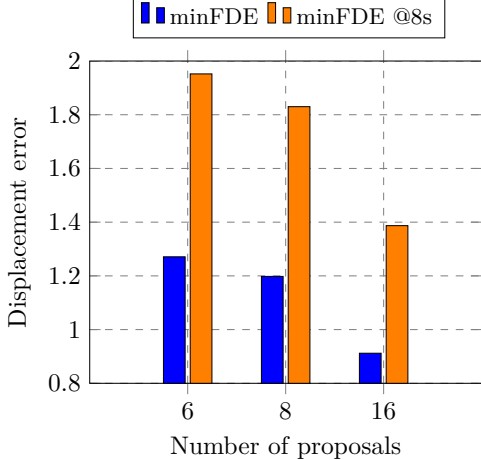

Figure 9: Increasing the number of predicted trajectory proposals

## D    Inference time

| Model (config) | #agents | Inference time |
|---|---|---|
| RedMotion (mlp-dec) | 1 | 23.2 ms ± 339 µs |
| | 8 | 25.2 ms ± 642 µs |
| | 64 | 87.9 ms ± 185 µs |
| RedMotion (tra-dec) | 1 | 18.2 ms ± 35.9 µs |
| | 8 | 29.7 ms ± 14.9 µs |
| | 64 | 146.0 ms ± 41.3 µs |

Table 3: **Inference times of our RedMotion models.** All times are measured on one A100 GPU using plain PyTorch eager execution and `torch.inference_mode()`. We show mean ± std. dev. of 7 runs with 10 loops each. #agents is the number of predicted agents.

# E  Challenge results in detail

| Object type | Measurement time (s) | Soft mAP | mAP | minADE | minFDE | Miss rate | Overlap rate |
|---|---|---|---|---|---|---|---|
| Vehicle | 3 | 0.5598 | 0.5416 | 0.2846 | 0.4999 | 0.0911 | 0.0183 |
| Vehicle | 5 | 0.4532 | 0.4468 | 0.5813 | 1.1248 | 0.1342 | 0.0411 |
| Vehicle | 8 | 0.3344 | 0.3321 | 1.1103 | 2.4313 | 0.2061 | 0.0964 |
| Vehicle | Avg | 0.4491 | 0.4402 | 0.6587 | 1.3520 | 0.1438 | 0.0519 |
| Pedestrian | 3 | 0.4541 | 0.4426 | 0.1650 | 0.3114 | 0.0644 | 0.2423 |
| Pedestrian | 5 | 0.3656 | 0.3608 | 0.3204 | 0.6504 | 0.0899 | 0.2681 |
| Pedestrian | 8 | 0.3128 | 0.3093 | 0.5665 | 1.2646 | 0.1237 | 0.2981 |
| Pedestrian | Avg | 0.3775 | 0.3709 | 0.3506 | 0.7422 | 0.0927 | 0.2695 |
| Cyclist | 3 | 0.4568 | 0.4487 | 0.3281 | 0.6089 | 0.1858 | 0.0510 |
| Cyclist | 5 | 0.3914 | 0.3883 | 0.6210 | 1.2206 | 0.2040 | 0.0888 |
| Cyclist | 8 | 0.2822 | 0.2803 | 1.0965 | 2.3765 | 0.2482 | 0.1394 |
| Cyclist | Avg | 0.3768 | 0.3725 | 0.6819 | 1.4020 | 0.2126 | 0.0930 |
| Avg | 3 | 0.4902 | 0.4776 | 0.2593 | 0.4734 | 0.1138 | 0.1039 |
| Avg | 5 | 0.4034 | 0.3987 | 0.5076 | 0.9986 | 0.1427 | 0.1327 |
| Avg | 8 | 0.3098 | 0.3072 | 0.9245 | 2.0241 | 0.1926 | 0.1780 |
| Avg | Avg | 0.4012 | 0.3945 | 0.5638 | 1.1654 | 0.1497 | 0.1382 |

Table 4: Detailed results of our RedMotion model (tra-dec configuration) on the test split of the Waymo Open Motion Challenge.

| Object type | Measurement time (s) | Soft mAP | mAP | minADE | minFDE | Miss rate | Overlap rate |
|---|---|---|---|---|---|---|---|
| Vehicle | 3 | 0.5641 | 0.5461 | 0.2852 | 0.5010 | 0.0915 | 0.0177 |
| Vehicle | 5 | 0.4567 | 0.4502 | 0.5813 | 1.1244 | 0.1342 | 0.0406 |
| Vehicle | 8 | 0.3316 | 0.3293 | 1.1128 | 2.4541 | 0.2078 | 0.0955 |
| Vehicle | Avg | 0.4508 | 0.4419 | 0.6598 | 1.3598 | 0.1445 | 0.0513 |
| Pedestrian | 3 | 0.4912 | 0.4773 | 0.1487 | 0.2726 | 0.0455 | 0.2370 |
| Pedestrian | 5 | 0.4025 | 0.3963 | 0.2830 | 0.5584 | 0.0655 | 0.2640 |
| Pedestrian | 8 | 0.3518 | 0.3462 | 0.4933 | 1.0499 | 0.0870 | 0.2951 |
| Pedestrian | Avg | 0.4152 | 0.4066 | 0.3083 | 0.6270 | 0.0660 | 0.2654 |
| Cyclist | 3 | 0.4581 | 0.4494 | 0.3318 | 0.6021 | 0.1826 | 0.0481 |
| Cyclist | 5 | 0.3660 | 0.3632 | 0.6213 | 1.2120 | 0.2064 | 0.0857 |
| Cyclist | 8 | 0.2678 | 0.2666 | 1.0954 | 2.4573 | 0.2593 | 0.1385 |
| Cyclist | Avg | 0.3640 | 0.3597 | 0.6829 | 1.4238 | 0.2161 | 0.0908 |
| Avg | 3 | 0.5045 | 0.4909 | 0.2552 | 0.4586 | 0.1065 | 0.1009 |
| Avg | 5 | 0.4084 | 0.4032 | 0.4952 | 0.9649 | 0.1354 | 0.1301 |
| Avg | 8 | 0.3171 | 0.3140 | 0.9005 | 1.9871 | 0.1847 | 0.1764 |
| Avg | Avg | 0.4100 | 0.4027 | 0.5503 | 1.1369 | 0.1422 | 0.1358 |

Table 5: Detailed results of our RedMotion model (tra-dec configuration) on the validation split of the Waymo Open Motion Challenge.

