# OpenReview forum: "RedMotion: Motion Prediction via Redundancy Reduction"
_TMLR — Accepted by TMLR_

### Review · Reviewer_346i · 2024-02-20

**Summary Of Contributions:**

The paper proposes an approach for motion prediction in autonomous driving, making use of self-supervised training, presented around the concept of redundancy reduction. Experiments show that the proposed self-supervised pre-training approach is efficient and the resulting method matches or outperforms strong prior baselines.

**Audience:**

Yes

**Broader Impact Concerns:**

No specific concerns

**Claims And Evidence:**

Yes

**Requested Changes:**

Address weaknesses 1-4 above.
1. Critical either to do it or explain well why it is not a good idea
2. Not 100% critical, but would make the paper stronger
3. Please explain this and also clarify in the paper
4. Please explain this and also clarify in the paper

**Strengths And Weaknesses:**

Pros:
1. Clear presentation, the paper is easy and pleasant to read
2. Reasonable method (however, I am not an expert in the motion prediction field and cannot really evaluate the novelty)
3. Good experiments, including some ablation studies
4. Good performance, better than or comparable to strong baselines
5. A colab allowing to try out the method is nice

Cons and questions:
1. Evaluation only on 1 dataset, that’s always somewhat suspicious, since maybe the method “overfitted” to the dataset at hand. What about Argoverse 2, as used by some of the baseline models?
2. It could be helpful to have more analysis/ablation experiments. For instance: on the effect of different data augmentations; on the effect of the amount of self-supervised data/training vs supervised; on the architectural details. The ablations/analysis that are already there are for sure helpful, but seem somewhat limited
3. Somewhat related to the previous, I was a bit confused by the following statement: “We use 100% of the training data for pre-training and fine-tune on only 12.5%”. Do the methods that are being compared to do this too? If not, does it mean that the proposed method uses substantially less labeled data? What happens when using 100% labeled data?
4. Some minor remarks and questions.
  a. What is the locality structure used in the local transformer? That is, how is it decided which tokens are neighbors with which for local attention?
  b. What exactly are the lines in Figure 4 middle two images?
  c. Typo: “ For our baseline model, our redundancy reduction reduction mechanism (b)”.

---

> ### Author Response · Authors · 2024-04-02
>
> Thank you for your feedback on our work. In response to common requests from reviewers, we have extended our experiments and updated our paper accordingly. In the new PDF version, significant changes are highlighted in yellow. Below are our responses to your specific questions.
>
> > Evaluation only on 1 dataset, that’s always somewhat suspicious, since maybe the method “overfitted” to the dataset at hand. What about Argoverse 2, as used by some of the baseline models?
>
> We performed additional experiments on the Argoverse 2 Forecasting dataset and added the results to Table 1. In detail, we compare our RedMotion model without pre-training versus with our mean-ag and env-traj pre-training configurations. Our pre-training methods improve both the minFDE and minADE scores. This shows that our pre-training methods improve the prediction accuracy on both datasets. Overall, the achieved displacement errors are higher for the Argoverse dataset than for the Waymo dataset, since the metrics are not averaged over multiple prediction horizons, and likely because the Argoverse training split is smaller (about 1M vs. 2M agent-centric samples).
>
> > I was a bit confused by the following statement: “We use 100% of the training data for pre-training and fine-tune on only 12.5%”. Do the methods that are being compared to do this too? If not, does it mean that the proposed method uses substantially less labeled data? What happens when using 100% labeled data?
>
> This is a slight misunderstanding. In our first set of experiments (Section 4.1), we pre-train all methods (our, PreTraM, Traj-MAE, and GraphDINO) on 100% of the datasets and fine-tune them on only 12.5\%. However, in Section 4.2, we also fine tune on 100% of the dataset. Thus, we always compare methods trained on the same amount of data. See Section 4.1 Datasets and Section 4.2 Dataset. To further highlight this, we have added this information to the caption of Table 1.
>
> > What is the locality structure used in the local transformer? That is, how is it decided which tokens are neighbors with which for local attention?
>
> We use local attention with a window size of 16 per layer. The locality structure is based on our sorting of the input tokens. As input to the road environment encoder, we use a list of these tokens sorted by token type, polyline, and distance to the current agent (see Section 3.2).
>
> > What exactly are the lines in Figure 4 middle two images?
>
> It shows how a lane token can gather information from a larger number of surrounding tokens throughout our local encoder (i.e., how its receptive field grows in subsequent local attention layers).

---

### Review · Reviewer_3sNL · 2024-02-26

**Summary Of Contributions:**

The authors propose the use of redundancy reduction techniques for representation learning in the context of trajectory prediction in autonomous driving applications. The method proposes two redundancy reduction techniques: one reduces a variable-sized set of local road environment tokens into a fixed-sized global embedding and the second applies the redundancy reduction principle to embeddings generated from augmented views of road environments. The results of the proposed self-supervised architecture are compared to other representation learning and trajectory prediction architectures on the Waymo Open Dataset.

**Audience:**

Yes

**Broader Impact Concerns:**

The authors provide thoughtful consideration for the various limitations of the proposed method. They conduct extensive experiment to analyze its performance. In addition to my suggestions above, it would be good to talk about concerns like out-of-distribution examples that may be encountered in the real-world during system deployment [8-9].

[8] Wiederer, Julian, et al. "Joint Out-of-Distribution Detection and Uncertainty Estimation for Trajectory Prediction." International Conference on Intelligent Robots and Systems (IROS). IEEE, 2023.

[9] Itkina, Masha, and Mykel Kochenderfer. "Interpretable self-aware neural networks for robust trajectory prediction." Conference on Robot Learning. PMLR, 2023.

**Claims And Evidence:**

Yes

**Requested Changes:**

Please see the weaknesses section above for my list of requested changes.

**Strengths And Weaknesses:**

Strengths:
* The general idea of investigating representation learning ideas that have had success in vision-language foundation models in the context of autonomous driving is worthwhile.
* The paper is well-written and easy to read.
* The experiments were thoughtfully constructed and fairly extensive.
* The discussion regarding potential reasons for certain observed trends was insightful.
* The authors did a great job at being transparent with respect to the potential weaknesses of the proposed approach.
* The paper is detailed with respect to implementation and provides code for good reproducibility.

Weaknesses:
* The caption for Fig. 1 could be made more meaningful (e.g., it does not currently discuss the redundancy reduction illustrated components). There seems to be no label for the global road environment encoding.
* In the introduction, the motivation is not particularly clear for why the two particular embeddings were chosen for redundancy reduction. Why do the authors think that these choices would most lead to good performance?
* In the related works, a sentence at the end of each section describing how the proposed method addresses the highlighted weaknesses of prior work would be helpful.
* The related works could be made more complete with the following citations. Trajectron++ [1] is a fairly well cited trajectory prediction architecture that would be good to consider. Furthermore, in the space of self-supervised motion prediction approaches are occupancy grid map (OGM) prediction techniques [2-7]. These methods construct OGMs from LiDAR data and most of them avoid having to label any data (e.g., in terms of bounding boxes or predicted trajectories).

[1] Salzmann, Tim, et al. ``Trajectron++: Dynamically-feasible trajectory forecasting with heterogeneous data.'' ECCV, 2020.

[2] Itkina, Masha et al. "Dynamic environment prediction in urban scenes using recurrent representation learning." Intelligent Transportation Systems Conference (ITSC). IEEE, 2019.

[3] Toyungyernsub, Maneekwan et al. "Double-prong ConvLSTM for spatiotemporal occupancy prediction in dynamic environments." International Conference on Robotics and Automation (ICRA). IEEE, 2021.

[4] Lange, Bernard et al. "Attention Augmented ConvLSTM for Environment Prediction." International Conference on Intelligent Robots and Systems (IROS). IEEE, 2021.

[5] Lange, Bernard, Masha Itkina, and Mykel J. Kochenderfer. "LOPR: Latent Occupancy PRediction using Generative Models." arXiv, 2022.

[5] Mohajerin, Nima and Mohsen Rohani. "Multi-step prediction of occupancy grid maps with recurrent neural networks." Proceedings of the IEEE/CVF Conference on Computer Vision and Pattern Recognition, 2019.

[6] Thomas, Hugues, et al. "Learning Spatiotemporal Occupancy Grid Maps for Lifelong Navigation in Dynamic Scenes." International Conference on Robotics and Automation (ICRA). IEEE, 2022.

[7] Mahjourian, Reza, et al. "Occupancy flow fields for motion forecasting in autonomous driving." IEEE Robotics and Automation Letters 7, 2022.

* It was not clear until Sec. 3.1 that road environment tokens include both roads and agents. In Sec. 3.1, it was also not very clear what the difference is between road environment tokens and global road environment descriptors (RED). Making this more clear would be helpful.
* In Sec. 3.2, it was not clear how the speed for the agents is estimated. Furthermore, getting bounding box/agent information for the trajectory prediction algorithms requires labeled data. What is the labeled data that the proposed pre-training scheme avoids?
* It is not clear what constitutes as pre-training versus finetuning in terms of loss and architecture in Sec. 4.1
* In the results, there is discussion that part of the drop in performance is that because the proposed method predicts all the trajectory points at once, which results in better FDE than ADE performance as in recurrent models. However, ultimately, the performance of the entire trajectory prediction is important.
* Fig. 6 is a bit difficult to read. The colors of the predicted trajectories are too close to the colors of the ground truth trajectories. The overlap also results in the disappearance of the past trajectories. Additionally, a qualitative figure with a baseline comparison would be helpful to understand the qualitative differences between methods.
* Some of the captions, including that for Fig. 6, are missing the takeaway that the reader should get from the figure. The Fig. 6 caption also mentions the color for bicyclists, but I don't see any in the figure?
* In Table 1, why is the worst performing variant of the proposed method (mean-ag) used as comparison for the baselines?
* It would be prudent to consider standard error or some measure of statistical significance for the quantitative results.
* Overall, the quantitative results are not very performant, particularly in Table 2.

* The following is the non-exhaustive list of typos that I found:
  * Top of page 6: "We use the local as [a query vector] ... generate [key and value] matrices"
  * Top of page 6: "[a dashed line]".
  * "this will [affect] the minFDE score more".

---

> ### Author Response · Authors · 2024-04-02
>
> Thank you for your feedback on our work. In response to common requests from reviewers, we have extended our experiments and updated our paper accordingly. In the new PDF version, significant changes are highlighted in yellow. Below are our responses to your specific questions and suggestions.
>
> > The caption for Fig. 1 could be made more meaningful (e.g., it does not currently discuss the redundancy reduction illustrated components). There seems to be no label for the global road environment encoding.
>
> We added a short description of our redundancy reduction mechanisms.
>
> > In the introduction, the motivation is not particularly clear for why the two particular embeddings were chosen for redundancy reduction. Why do the authors think that these choices would most lead to good performance?
>
> Please refer to the second paragraph in our introduction:
> Specifically, our model learns augmentation-invariant features of road environments as self-supervised pre-training. We hypothesize that by using these features, relations in the road environment can be learned, providing important context for motion prediction.
>
> > The related works could be made more complete with the following citations.
>
> Thank you for the suggestions. We have included the methods Trajectron++ and HPTR in the related work section.
>
> > It was not clear until Sec. 3.1 that road environment tokens include both roads and agents.
>
> Please refer to the abstract: Our first type of redundancy reduction is induced by an internal transformer decoder and reduces a variable-sized set of local road environment tokens, representing road graphs and agent data, to a fixed-sized global embedding.
>
> > Getting bounding box/agent information for the trajectory prediction algorithms requires labeled data. What is the labeled data that the proposed pre-training scheme avoids?
>
> For all our configurations besides the env-traj configuration, we exclude the past trajectory and use only the current state of surrounding agents. Therefore, our method requires significantly less annotated data than the closely related methods PreTraM and TrajMAE (see Section 4.1).
>
> > It is not clear what constitutes as pre-training versus finetuning in terms of loss and architecture in Sec. 4.1
>
> We pre-trained both our baseline (without architecture-induced redundancy reduction (a)) and our proposed RedMotion model with the mean-ag configuration of our self-supervised redundancy reduction. Please refer to the difference in relative performance improvements in Table 1 or Section 4.1.
>
> > In the results, there is discussion that part of the drop in performance is that because the proposed method predicts all the trajectory points at once, which results in better FDE than ADE performance as in recurrent models. However, ultimately, the performance of the entire trajectory prediction is important.
>
> We agree, this applies to our model with an MLP-based head. Therefore, we additionally trained a version of our model with a transformer decoder as motion head, which improves the mAP and minADE scores. Please refer to the second paragraph in Section 4.2.
>
> > Why is the worst performing variant of the proposed method (mean-ag) used as comparison for the baselines?
>
> We compare all of our configurations to related methods, and the relative performance improvements of the first 11 rows in Table 1 are computed with respect to the baseline model without any pre-training. However, we only evaluated our mean-ag pre-training configuration with "only" the baseline model and the remaining configurations in combination with our architecture-induced redundancy reduction mechanism (i.e., using our proposed RedMotion model). This shows that our pre-training also improves the performance of a baseline model, but works best in combination with our architecture-induced redundancy reduction.
>
> > It would be prudent to consider standard error or some measure of statistical significance for the quantitative results.
>
> We extended our experiments: trained models on the Argoverse 2 Forecasting dataset and report mean and std. dev. of 3 replicas per method and configuration (see Table 1).
>
> > Overall, the quantitative results are not very performant, particularly in Table 2.
>
> We respectfully disagree since our method outperforms all comparable methods in the main challenge metric test Soft mAP.
>
> >  It would be good to talk about concerns like out-of-distribution examples that may be encountered in the real-world during system deployment.
>
> We have added a note on robustness w.r.t. OOD samples to the limitations section.

---

### Review · Reviewer_6Nx1 · 2024-03-19

**Summary Of Contributions:**

This paper presents RedMotion, a transformer based motion-prediction model. RedMotion uses self-supervised learning to learn joint representations of augmented and non-augmented road features - akin to vision methods like SimCLR. Additionally, RedMotion embeds local features into a fixed-size global embedding for more compressed representations. All experiments are conducted on the Waymo Motion-Prediction Challenge where models are evaluated on displacement metrics (with respect to ground-truth trajectories). RedMotion achieves near SoTA or slightly worse performance than prior methods despite being fine-tuned only on 12% of the data.

**Audience:**

Yes

**Broader Impact Concerns:**

Nothing of immediate concern.

**Claims And Evidence:**

Yes

**Requested Changes:**

Based on the comments above:
- Add ablation and sensitive analyses to show the importance of individual components.
- Do online evaluations to show usefulness in realistic driving scenarios.
- Explain the overfitting to validation data.
- Fine-tune with more data: 40-100% mixtures
- Discuss the difference to vision-based self-supervised techniques that use raw data.

**Strengths And Weaknesses:**

**Strengths**
+ RedMotion achieves compelling performance while only using 12% of the data for fine-tuning. The motion-prediction challenge contains challenging real-world driving scenarios.
+ The authors provide a Colab notebook with the implementation. The method and results should be easily reproducible with this setup.
+ The experiments include a good set of baselines. RedMotion is benchmarked against Traj-MAE, PreTraM, MotionCNN, SceneTransformer etc.
+ The methods and experimental setup are mostly clear and easy to follow.



**Weaknesses**
- The experiments are missing key ablations and sensitivity experiments. How sensitive are the results to augmentation parameters like perturbation translation and rotation? Were the two types of reductions, i.e.  (1) local to global tokens, (2) data augmentation, ablated to show how much each reduction adds to the performance?
- The motion-prediction models are only evaluated with offline displacement metrics and not with online evaluations. Displacement metrics might not directly correlate with usefulness for self-driving scenarios as trajectories are often multi-modal; there could be several valid trajectories given a start state. It would be interesting to use RedMotion prediction in a simulated self-driving setting to see if it helps improve performance or in providing new capabilities.
- The validation results for RedMotion seem much better than test results. Does this indicate overfitting? This seems less of a problem for other methods like SceneTransformer. Perhaps fine-tuning only on 12% of the data is prone to overfitting as the tuning process is over-reliant on the validation set.
- Since the overall performance is near or worse-than SoTA methods, why not fine-tune on more than 12% of the data? Perhaps adding more variants like 24%, 60%, 80%, and 100% would provide insights into the strengths and limitations of the pre-training approach.
- Even though the paper focuses on “self-supervised” learning, RedMotion relies on a lot of annotated road features which are expensive to label. Traditional self-supervised methods in vision like SimCLR learn representations just from images. The equivalent here would be to learn directly from raw point-cloud or image data. So, RedMotion’s approach is not particularly scalable.

---

> ### Author Response · Authors · 2024-04-02
>
> Thank you for your feedback on our work. In response to common requests from reviewers, we have extended our experiments and updated our paper accordingly. In the new PDF version, significant changes are highlighted in yellow. Below are our responses to your specific questions.
>
> > Were the two types of reductions, i.e. (1) local to global tokens, (2) data augmentation, ablated to show how much each reduction adds to the performance?
>
> Yes, we pre-trained both our baseline model (without architecture-induced redundancy reduction) and our proposed RedMotion model with the mean-ag configuration of our self-supervised redundancy reduction. Please refer to the difference in relative performance improvements in Table 1 or Section 4.1.
>
> > The experiments are missing key ablations and sensitivity experiments. How sensitive are the results to augmentation parameters like perturbation translation and rotation?
>
> We respectfully disagree since we follow common practice in self-supervised learning (e.g., BYOL, VICReg, Barlow Twins) by using a fixed set of reasonable augmentations. Thereby, we avoid giving our method an advantage over related methods by fine-tuning the used augmentations to our method.
>
> > The motion-prediction models are only evaluated with offline displacement metrics and not with online evaluations. It would be interesting to use RedMotion prediction in a simulated self-driving setting.
>
> We agree that research in motion prediction should focus on methods that are real-time capable and therefore usable in self-driving stacks. Therefore, we developed RedMotion without using ensembling or trajectory aggregation methods. See Table 2, where our model outperforms all other related methods that meet these requirements (main challenge metric test Soft mAP), and Appendix D, which shows the inference latency of our method. However, an evaluation in an end2end fashion (as part of a complete self-driving system) is beyond the usual scope of motion prediction papers (e.g., HPTR, Wayformer, MTR).
>
> > The validation results for RedMotion seem much better than test results. Does this indicate overfitting?
>
> Yes, the prediction accuracy of our method is higher on the validation set than on the test set. However, as you noted, this is also true for Scene Transformer. Furthermore, we have included the validation performance of the newer method HPTR in the new version of our paper. For HPTR, the performance difference between validation and test split is larger than for our method. Therefore, we assume that the validation and training sets are more similar than the training and test sets. Furthermore, some samples in the test set do not contain map data, which could worsen the performance of our method, which specifically uses map data for pre-training (see the [dataloader](https://github.com/zhejz/HPTR/blob/646e07084da8049bc3d8f682b7378f640225b258/src/pack_h5_womd.py#L366) of HPTR).
>
> > Since the overall performance is near or worse-than SoTA methods, why not fine-tune on more than 12% of the data?
>
> This is a slight misunderstanding. In our first set of experiments (Section 4.1), we pre-train all methods (our, PreTraM, Traj-MAE, and GraphDINO) on 100% of the datasets and fine-tune them on only 12.5%. However, in Section 4.2, we also fine tune on 100% of the dataset. Thus, we always compare methods trained on the same amount of data. See Section 4.1 Datasets and Section 4.2 Dataset. To further highlight this, we have added this information to the caption of Table 1.
>
> > Even though the paper focuses on “self-supervised” learning, RedMotion relies on a lot of annotated road features which are expensive to label. Traditional self-supervised methods in vision like SimCLR learn representations just from images.
>
> We agree that self-supervised methods in computer vision like SimCLR use less annotated data than we do in our method.
> However, “how much annotated data is allowed in self-supervised learning?” or “how to define annotated data in self-supervised learning?” are open research questions. For example, CLIP or GPT models use text as (self-)supervision signal during pre-training, while text can also be considered as a label in the corresponding downstream tasks (image captioning and general purpose language modeling). ForecastMAE uses the future trajectory to generate self-supervision signals for motion prediction. We draw the line at the downstream labels (for motion prediction) and thus strictly exclude the future trajectory from our pre-training. For all our configurations besides the env-traj configuration, we also exclude the past trajectory and use only the current state of surrounding agents. Therefore, our method requires significantly less annotated data than the closely related methods PreTraM and TrajMAE (see Section 4.1). Finally, our main focus is on redundancy reduction: architecture-induced (a) and self-supervised (b) rather than just self-supervised.

---

### Decision · Action_Editor_ee7c · 2024-04-24

**Recommendation:** Accept as is

**Comment:**

All reviewers lean towards acceptance of the manuscript and that it is well written work.  Note, they do list several important points (particularly around ablations) that would strengthen the impact of the work.

**Audience:**

Autonomous driving researchers

**Claims And Evidence:**

The paper presents a self-supervised transformer for motion-prediction evaluated on the Waymo Motion-Prediction Challenge and are data efficient -- requiring only a small amount of fine-tuning data.  The additional insight is redundancy reduction which provides a sort of bottleneck in the representation.